# Brain Noradrenergic Innervation Supports the Development of Parkinson’s Tremor: A Study in a Reserpinized Rat Model

**DOI:** 10.3390/cells12212529

**Published:** 2023-10-27

**Authors:** Nicoló Gabriele Pozzi, Francesco Bolzoni, Gabriele Eliseo Mario Biella, Gianni Pezzoli, Chi Wang Ip, Jens Volkmann, Paolo Cavallari, Esther Asan, Ioannis Ugo Isaias

**Affiliations:** 1Department of Neurology, University Hospital and Julius-Maximilians-Universität Würzburg, Josef-Schneider-Str. 11, 97080 Würzburg, Germany; pozzi_n2@ukw.de (N.G.P.); ip_c@ukw.de (C.W.I.); volkmann_j@ukw.de (J.V.); 2Department of Biomedical Sciences, Humanitas University, Pieve Emanuele, 20072 Milano, Italy; francesco.bolzoni@unimi.it; 3Institute of Molecular Bioimaging and Physiology, CNR, Via Fratelli Cervi 93, 20090 Milano, Italy; gembiella@gmail.com; 4Centro Parkinson e Parkinsonismi, ASST G. Pini-CTO, 20072 Milano, Italy; pezzoli@parkinson.it; 5Department of Pathophysiology and Transplantation, Human Physiology Section, Università degli Studi di Milano, via Mangiagalli 32, 20133 Milano, Italy; paolo.cavallari@unimi.it; 6Institute of Anatomy and Cell Biology, Julius-Maximilians-Universität Würzburg, Koellikerstr 6, 97070 Würzburg, Germany; esther.asan@uni-wuerzburg.de

**Keywords:** Parkinson’s disease, tremor, locus coeruleus, noradrenaline, reserpinized rat model

## Abstract

The pathophysiology of tremor in Parkinson’s disease (PD) is evolving towards a complex alteration to monoaminergic innervation, and increasing evidence suggests a key role of the locus coeruleus noradrenergic system (LC-NA). However, the difficulties in imaging LC-NA in patients challenge its direct investigation. To this end, we studied the development of tremor in a reserpinized rat model of PD, with or without a selective lesioning of LC-NA innervation with the neurotoxin DSP-4. Eight male rats (Sprague Dawley) received DSP-4 (50 mg/kg) two weeks prior to reserpine injection (10 mg/kg) (DR-group), while seven male animals received only reserpine treatment (R-group). Tremor, rigidity, hypokinesia, postural flexion and postural immobility were scored before and after 20, 40, 60, 80, 120 and 180 min of reserpine injection. Tremor was assessed visually and with accelerometers. The injection of DSP-4 induced a severe reduction in LC-NA terminal axons (DR-group: 0.024 ± 0.01 vs. R-group: 0.27 ± 0.04 axons/um^2^, *p* < 0.001) and was associated with significantly less tremor, as compared to the R-group (peak tremor score, DR-group: 0.5 ± 0.8 vs. R-group: 1.6 ± 0.5; *p* < 0.01). Kinematic measurement confirmed the clinical data (tremor consistency (% of tremor during 180 s recording), DR-group: 37.9 ± 35.8 vs. R-group: 69.3 ± 29.6; *p* < 0.05). Akinetic–rigid symptoms did not differ between the DR- and R-groups. Our results provide preliminary causal evidence for a critical role of LC-NA innervation in the development of PD tremor and foster the development of targeted therapies for PD patients.

## 1. Introduction

Tremor is a cardinal motor symptom of Parkinson’s disease (PD), present in up to 75% of patients [1], and a severe cause of disability and social stigma [2]. The pathophysiology of PD tremor remains largely unclear [3] and should account for several distinctive features [3,4]. Unlike akinetic–rigid symptoms, PD tremor poorly correlates with striatal dopaminergic loss, the hallmark of PD pathophysiology [5,6]; it can be resistant to dopaminergic treatments [7,8], it may disappear along with disease progression [1], it increases in amplitude or is triggered by psychological states of anxiety or stress [9,10] and it ceases during sleep. These characteristics lead to the hypothesis that the pathophysiology of PD tremor extends beyond dopamine denervation and comprises alterations in other monoaminergic neurotransmitters [3,4]. Recent findings suggested a role for acetylcholine, serotonin and noradrenalin in tremor development, with different neurotransmitter interactions determining peculiar tremor characteristics, such as re-emergent and pure postural tremor (see, for review [3,4]). With regard to tremor development, converging evidence points to the noradrenergic system [4,11,12]. Pathological studies showed that tremor-dominant PD patients have a relatively preserved locus coeruleus (LC) [13], which represents the primary source of noradrenaline (NA) in the brain [14]. Accordingly, we described a specific metabolic network in tremor-dominant PD patients that comprised the cerebellum, the primary motor cortex, the putamen and an area in the dorsal pons, which corresponds to the LC [15]. A subsequent molecular imaging study confirmed that LC-NA innervation is relatively preserved in tremor-dominant PD patients [16]. In line with this, intravenous injection of adrenaline was shown to increase tremor in patients with PD [17]. An EMG-fMRI study also recently showed that increased tremor in PD under cognitive stress is associated with the activation of the ascending arousal system, which includes NA terminals [10].

The study of LC-NA innervation in PD patients is still challenged by the many difficulties in imaging LC activity in humans, and the current evidence remains correlative [4]. Longitudinal investigations are difficult because it is virtually impossible to control for the degenerative process in PD, which likely affects the dopamine and LC-NA system differently [13]. The follow-up of subjects at risk of developing PD, such as subjects with REM sleep behaviour disorder, is also limited by the alteration to LC-NA in these subjects [18].

To avoid these limitations and directly assess the role of LC-NA innervation in PD tremor, we studied the development of tremor in the reserpinized rat model of PD [19,20,21], with and without selective lesioning of the LC-NA terminals using N-(2-Chloroethyl)-N-ethyl-2-bromobenzylamine (DSP-4) [22]. Comparing animals with and without LC-NA denervation, we showed that LC-NA plays a pivotal role in the development of PD tremor.

## 2. Materials and Methods

### 2.1. Animals

We investigated 15 male Sprague Dawley rats (Charles River Laboratories, Sulzfeld, Germany) constantly kept under standard conditions (21 °C, 12 h light/dark cycle). Food and water were available ad libitum. At the time of the experiment, animals weighed between 200 and 220 g. Experiments were carried out one week post-arrival at a fixed time (between 3 p.m. and 5 p.m.; light was on at 8 a.m.) and took place in a dedicated room in the animal facility of the IBFM, CNR, Segrate (MI), Italy, after 30 min of acclimatization. All animals investigated in this study were handled according to applicable international, national and institutional guidelines for care and use of animals, and all efforts were made to minimize animal suffering. The local institutional review board approved the experiments.

### 2.2. Drug Treatment Procedures and Dose Selection

We opted for the reserpinized rat model of PD, one of the very few that presents tremor [19,20,21]. Reserpine is an irreversible inhibitor of vesicular monoamine transporter 2 (VMAT-2) [23]. The disruption of VMAT-2 depletes intracellular monoamine storage, and the consequent reduction in monoamines in nerve terminals causes bradykinesia, rigidity and tremor [19,24,25,26]. The additional inhibition of amine uptake further leads to the accumulation of neurotoxic oxidation byproducts [27], which results in neuronal damage [28,29].

The selective LC-NA damage was achieved via a pre-treatment with toxin DSP-4, which was applied 14 days prior to reserpine injection in a group animal. The systemic injection of DSP-4 causes a depletion in NA levels, in the release capacity and in the activity of dopamine beta-hydroxylase (DBH) [30]. In the first two weeks after treatment, the neurotoxin exclusively affects and destroys NA terminal axons arising from the LC [30,31] because of the specific binding proprieties of the NA transporter in LC axon terminals that maximize DSP-4 affinity and uptake, leading to local alkylation of vital proteins [32].

According to treatment, animals were divided into two groups. The first group of seven rats (R-group) received only one intraperitoneal injection of 10 mg/kg of reserpine [33]. Instead, the second group of eight rats (DR-group) was pre-treated with intraperitoneal injection of 50 mg/kg of DSP-4. To ensure sufficient degeneration of LC-NA axon terminals prior to experiment, the reserpine treatment was performed 14 days after DSP-4 injection [34].

### 2.3. Behavioral Evaluations and Kinematic Analysis

Observations were carried out before the reserpine injection and for the following 3 h. All animals were sacrificed just after the end of the experiments.

We used the visual 0–2 points scoring system [19]: score 2 was assigned when tremulous movements were visible immediately and clearly, score 1 when tremulous movements were intermittent and of modest amplitude and score 0 when no tremulous movements could be observed. This scoring system was also used for rigidity, hypokinesia, postural flexion of the back and postural immobility. Of note, we did not investigate DR animals with tests specifically engaging the noradrenergic system (e.g., novelty seeking, learning, etc.) as we focused our observations on PD motor symptoms.

All evaluations were carried out by a single examiner, who was unaware of treatment conditions, before reserpine injection (0 min) and 20, 40, 60, 80, 120 and 180 min after reserpine injection. Maneuvers were repeated three times in total, and the average score for each animal was calculated.

We also assessed the kinematics of tremor with accelerometers attached to the back limbs of the rat. Tremor was measured as the variation in the acceleration of the most tremulous limb. We computed the consistency of tremor (T%) as the percentage of the total time recorded (average of three sessions of ≈60 s at each time point following reserpine injection). Recordings from all the animals but one (r4, R-group) were available and were analyzed with Matlab-based (Mathworks) custom scripts. Kinematic data from r4 were excluded due to the presence of artifacts, which were visually inspected for all recordings.

### 2.4. Tissue Preparation, Immunolabeling and Quantitative Analyses

Immediately after the end of the recordings, the rats were sacrificed by inducing a deep anesthesia level using isoflurane in combination with a high-dosage injection of barbiturate (pentobarbital). Once the death was ensured, transcardial perfusion with cold heparinized saline solution was performed. Tissues were fixed through perfusion with 4% solution of paraformaldehyde in phosphate-buffered saline (PBS). The brain was then removed and preserved in 4% paraformaldehyde until the histological analyses were performed.

Tissue preparation, sectioning and immunofluorescence labeling were carried out as previously described [35,36]. Briefly, tissue blocks containing upper pons and cerebellum were washed in 0.01 M PBS (pH 7.4), successively infiltrated with 10 and 20% sucrose in PBS, frozen in liquid-nitrogen-cooled isopentane and stored at −80 °C. Serial 40 µm frontal vibratome sections were prepared after thawing the tissue to room temperature (RT). Preincubation of free-floating sections in 5% normal goat serum (NGS; Sigma, Deisenhofen, Germany) and 1% Triton X-100 (TX100; Sigma) in PBS for 2 h at RT was followed by incubation in the primary antibody solution for 48–72 h at 4 °C. Antibodies used were polyclonal rabbit-anti-tyrosine hydroxylase (TH; Millipore, Schwalbach, Germany) and polyclonal rabbit-anti-dopamine-β-hydroxylase (DBH, Abcam, Cambridge, UK), diluted 1:500 in 1% NGS, 0.5% TX100 and 0.05% NaN_3_ in PBS. After washing in PBS, sections were incubated in Cy3-labeled goat-anti-rabbit secondary antibody (Dianova, Eiching, Germany, 1:600) in 0.5% TX100 in PBS for 2 h at RT, washed in PBS, mounted on Superfrost^TM^ microscopic slides and coverslipped with Fluorogel (Electron Microscopy Sciences, Munich, Germany). Microphotographs for images were taken with a Keyence BZ 9000 microscope. Control sections subjected to the reaction sequence without primary antibodies did not show specific labeling.

Quantification of TH and DBH neurons and axon densities was carried out with a Zeiss Axiophot2 microscope using digital images acquired via CCD camera and ImagePro 4.0 software. We focused on the LC and the stratum granulosum (SG) and stratum moleculare (SM) of cerebellar vermal cortical areas, due to the accessibility of the region and the documented role of the cerebellum in tremor [37,38,39]. For axon density assessment, six images in the region showing the locus coeruleus were analyzed in two sections per animal. Each structure within the molecular layer of the first lobule of the upper portion of the cerebellar vermis that demonstrated a continuous TH+ immunoreactive fiber profile was counted as one axon. For the neuron count, cell profiles displaying TH+ reactivity within the LC were quantified on two sections per animal. Axonal and neuronal densities per µm^2^ were then calculated after determination of the analyzed cortical area and the area of the LC, respectively, with ImageJ2.

## 3. Statistical Analysis

Statistical analyses were performed with the JMP statistical package (version 13, SAS Institute, Inc., Cary, NC, USA). Data are presented as mean ± standard deviation (SD). After testing for normality distribution of the data, we assessed statistical significance by means of sample *t*-test for the assessment of noradrenergic denervation between DR- and R-groups. Due to the ordinal variables of the behavioral data, these were assessed by means of Mann–Whitney U test. Finally, differences in tremor consistency over time between groups were studied by means of two-way mixed model ANOVA and Wilcoxon sign-rank test. The threshold level of statistical significance was set for every analysis at *p* < 0.05.

## 4. Results

### 4.1. Assessment of DSP-4-Induced Noradrenergic Denervation

We showed a marked reduction in LC-NA axon terminals in the DR-group as compared to the R-group. Reserpine treatment alone did not affect noradrenergic axons reaching the cerebellar cortex, as shown by the TH (Figure 1A) and DBH immunolabeling (Figure 1B) of one R animal. In contrast, the pre-treatment with DSP-4 led to a profound loss of TH- and DBH-immunolabeled axonal profiles in the cerebellar cortex (Figure 1C,D).

Quantitative analyses confirmed a ~90% reduction in the density of TH+ noradrenergic axon profiles in the DR-group (0.024 ± 0.01 axons/µm^2^) compared to the R-group (0.270 ± 0.04 axons/µm^2^; sample *t*-test, *p* < 0.001), providing evidence that DSP-4 caused a significant LC-NA denervation. On the contrary, the cell density of TH+ noradrenergic perikarya of the LC did not differ between the groups, with an average cell density of 1.37 neurons/µm^2^ in DR animals and 1.33 neurons/µm^2^ in R animals (*p* > 0.05), thus supporting a selective effect of DSP-4 on LC-NA axon terminals [30]. The results are displayed in Figure 2.

### 4.2. Visual Scores

Behavioral evaluations prior to reserpine injection did not show any motor symptoms in either group. After the injection of reserpine, DR animals had significantly less tremor compared with R animals (score at 40 min: 0.50 ± 0.76 vs. 1.57 ± 0.53, respectively, Mann–Whitney U test, *p* < 0.01; score at 60 min: 0.12 ± 0.35 vs. 1.14 ± 0.90, *p* < 0.01) (Figure 3). Instead, sustained rigidity, hypokinesia, postural flexion of the back and postural immobility were documented in both groups (Figure 3). These akinetic–rigid symptoms did not differ at any time point between the DR- and R-groups (Figure 3). Tremor peaked early (i.e., 40 min) and decreased over time, vanishing after 120 min in all animals. Akinetic–rigid symptoms peaked late (i.e., 60–80 min) and did not diminish (Figure 3).

### 4.3. Accelerometer Tremor Measurements

Kinematic measurement mirrored the visual scoring and showed a significant difference in the consistency of tremor between the DR- and R-groups at all time points (Figure 3). Tremor was marginally present in the DR-group as compared to the R-group, with an average T% of 23.98 ± 28.45 s vs. 45.46 ± 35.66 s, respectively (Wilcoxon sign-rank test, *p* = 0.01). Measurements before and after 40, 60 or 80 min from reserpine injection did not record any tremor.

## 5. Discussion

In this study, we showed that depletion in cerebellar noradrenergic innervation reduced the development of tremor in the reserpinized rat model of PD (Figure 3). This finding provides preliminary causal evidence for the role of LC-NA in the development of PD tremor.

In PD, dopamine depletion can cause the entrainment of the cerebellar–thalamic–cortical loop in pathological rhythms [37,40], and preserved LC-NA activity could facilitate the emergence of tremor via cerebellar excitatory (glutamatergic) output to the motor thalamic nuclei [4,41]. The LC receives multiple varied inputs that can increase LC-NA firing to differing extents [42] and, therefore, account for the different life conditions inducing or enhancing PD tremor [9], such as challenging cognitive tasks [10].

In line with this, we previously showed that LC-NA reuptake is up-regulated in early PD patients [43], and a recent molecular imaging study with 11C-MeNER, a reboxetine analogue that binds specifically to LC-NA terminals, found a relatively preserved LC-NA innervation in tremor-dominant PD patients, as compared to akinetic–rigid PD patients [16]. The uptake of 11C-MeNER in the LC of tremor-dominant PD patients was comparable to healthy controls and positively correlated with tremor severity [16]. By combining 11C-MeNER imaging and neuromelanin-sensitive MRI, which is sensitive to the loss of LC neurons, it was also shown that LC-NA axonal damage exceeded somatic damage in PD [44]. These findings closely mirror the degeneration pattern that we achieved with DSP4, which induced an extensive reduction in LC-NA terminals while largely sparing the LC cell bodies (Figure 1), as assessed with TH-immunoreactivity analysis (Figure 2). This peculiar degeneration pattern is likely related to the extremely large arborization of NA neurons [45], which makes them particularly vulnerable to toxic insults. This may be responsible for a relatively rapid degeneration of LC-NA innervation that may, in turn, explain the reduction in tremor along with PD progression [41].

In our study, we also showed that in reserpinized animals, tremor peaks early and vanishes over time (Figure 3), while akinetic–rigid symptoms progressively increase [5,6]. These distinctive patterns closely resemble the natural evolution of PD [5,6], thus supporting the translational values of our findings and the idea of different mechanisms for PD tremor and akinetic–rigid symptoms, with the former being triggered by LC-NA activity and the latter correlating with dopamine depletion [4,5,6]. The selective reduction in tremor over time also follows the mechanism of reserpine neural damage [19,24,25,26] and reflects the depletion of intracellular NA storage.

Our study has some limitations. First, we used a toxic animal model of PD [19,20,21], not reflecting the chronic progressive neurodegeneration of this neurological disorder. However, this model was the best suited for investigating tremor. Despite being an acute model, the reserpinized rat is one of the very few showing tremor along with akinetic–rigid symptoms [19,20,21]. The widespread effect of reserpine represents a second limitation in that it affects all monoamines, also resulting in serotonin depletion. A reduction in raphe serotonin was shown to correlate with tremor severity in a recent molecular imaging study from the Parkinson’s progression markers initiative [46]. While we cannot rule out an effect of serotonin depletion on tremor, the striking difference induced by the selective lesioning of NA-LC with DSP-4 neurotoxin supports a pivotal role for LC-NA. Of note, studies on DSP-4 injections in rats showed that this neurotoxin is highly selective for LC-NA terminals and induces only a marginal reduction in serotoninergic levels [47]. Yet, we must acknowledge that we limited the quantification of noradrenergic innervation to the analysis of TH and DBH neurons and axon densities under fluorescence microscopy. Although this was easily achieved because of the clear differentiation of the fibers on visual examination, the lack of immunohistochemistry with of nuclear counterstaining with DAB (3, 3′-diaminobenzidine) and DAPI blue (4′,6-diamidino-2-phenylindole) limits the precision in quantifying cell bodies and axons. Furthermore, we did not include a control group with DSP4 treatment only; thus, our findings must be interpreted with caution. The third limitation is the selective investigation of cerebellar noradrenergic innervation, which further limits our reasoning on the relevance of other structures of the cerebellar–thalamic–cortical loop in the development of PD tremor. This choice was made because of the well-established role of the cerebellum in the origin of tremor [48]. Future studies will need to describe the noradrenergic impact on each structure of the cerebellar–thalamic–cortical loop and the basal ganglia for PD tremor and for other motor and non-motor symptoms of PD. With converging evidence pointing to a combined alteration in noradrenergic, acetylcholinergic and serotoninergic transmission at the origin of tremor subtypes (e.g., resting and postural components) [3,7,8,11], future investigation might lead to the development of targeted treatments.

## 6. Conclusions

Our findings support a pivotal role of brain noradrenergic transmission in the development of PD tremor. The involvement of the LC-NA system may also explain many of its clinical features, such as the intermittent course, worsening under cognitive stress and suboptimal response to dopaminergic replacement therapy. The development of new therapeutic approaches also targeting the noradrenergic system is necessary for better tremor control in patients with PD.

## Figures and Tables

**Figure 1 cells-12-02529-f001:**
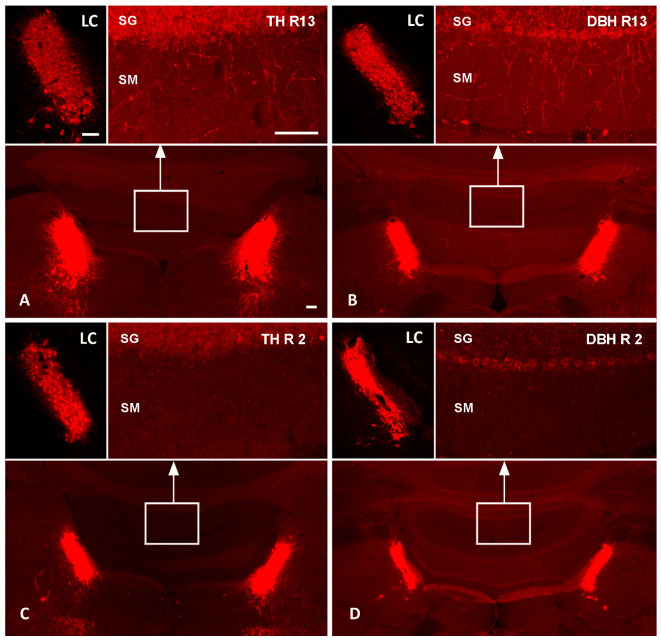
Effects of DSP-4 on noradrenergic locus coeruleus neurons and their terminal axons in the cerebellum. TH and DBH fluorescence immunolabeling of the pontine brainstem and cerebellum of one R-group animal (R13; (**A**,**B**)) and one DR-group (R2; (**C**,**D**)) in the top and bottom row, respectively. The top-left images in each panel show higher magnifications of the left LCs; the top-right images show higher magnifications of the stratum granulosum (SG) and stratum moleculare (SM) of cerebellar vermal cortical areas indicated by white boxes in the overviews. In the reserpinized rat (**top row**), TH (**A**) and DBH immunoreactions (**B**) label noradrenergic neurons in the LC and numerous noradrenergic terminal axons in the cerebellar cortex. In the DSP-4-treated animal (**bottom row**), TH (**C**) and DBH immunoreactions (**D**) document a severe loss of cerebellar noradrenergic terminal axons, while the LC neuronal cell bodies appear relatively spared. Bars: 100 µm. DBH, dopamine beta-hydroxylase; LC, locus coeruleus; TH, tyrosine hydroxylase.

**Figure 2 cells-12-02529-f002:**
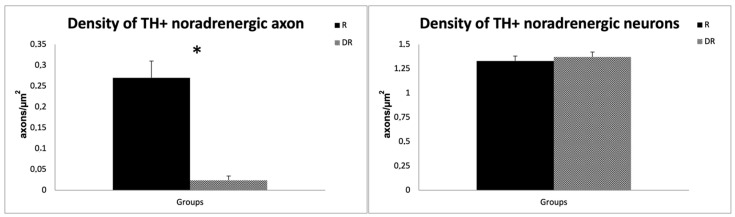
Quantitative analysis of DSP-4 on TH-labelled axons and neurons. The left graph shows the difference in axon density for TH-labeled noradrenergic neurons in R- and DR-groups. The right graph displays the difference in perikarya for TH-labeled noradrenergic neurons. R animals are shown in black, DR-animals in grey. DSP-4: N-(2-Chloroethyl)-N-ethyl-2-bromobenzylamine; TH, tyrosine hydroxylase. * *p* < 0.05.

**Figure 3 cells-12-02529-f003:**
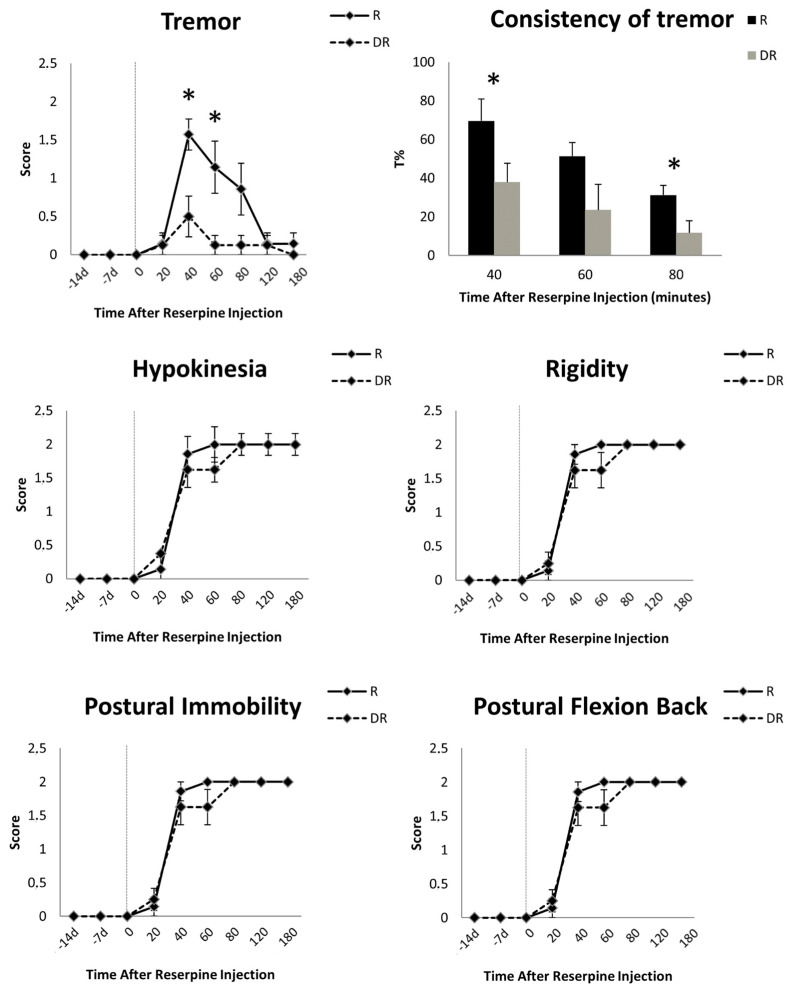
Time course of the motor signs induced by 10 mg/kg of reserpine in rats. Data are shown as mean (±standard error of mean, SEM) of seven reserpinized-only animals (R-group) and eight animals pre-treated with DSP-4 two weeks before reserpine (DR-group). For these animals only, the motor effects of DSP-4 are reported at 14 and 7 days before reserpine injection. Asterisks indicate statistical significance. Tremor severity and consistency (T%, i.e., the percentage of the total time with tremor as recorded with an accelerometer placed on the most tremulous limb) differed between groups, being more represented in the R-group. This difference was not mirrored by akinetic–rigid symptoms, which were equally present and severe in the R- and DR-groups.

## Data Availability

The datasets used and analyzed during the current study are available from the corresponding author upon request. Inquires can be sent to the corresponding author.

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
