# Peer review of "Brain Noradrenergic Innervation Supports the Development of Parkinson’s Tremor: A Study in a Reserpinized Rat Model"

_cells, 2023, doi:10.3390/cells12212529_

Round 1

Reviewer 1 Report

The study by Pozzi et al. addresses the knowledge gap on the origins of tremors in PD. Therefore, this manuscript directly explores the role of LC-NA innervation in PD tremors. The researchers examined tremor development in a rat model of PD with and without selective LC-NA terminal lesioning using DSP-4. Overall, the study demonstrated that DSP-4 treatment led to a marked reduction in noradrenergic axon terminals originating from the LC-NA system, confirming a significant LC-NA denervation. This reduction in axon terminals was specific and did not affect the cell bodies of LC neurons. The behavioral effects were consistent with the reduction in tremors observed in the DSP-4 treated group, while akinetic-rigid symptoms were not affected by the treatment. By comparing animals with intact and denervated LC-NA systems, the authors concluded that LC-NA innervation plays a crucial role in developing PD tremors.

Major Comment:

Figure 1 does not present the quantitative analysis data mentioned in the result section. The authors need to include this graph to corroborate the visual findings.

Author Response

We thank the reviewer for the suggestion. We added a graph/figure (Figure 2) displaying the difference between groups to corroborate the findings of Figure 1.

Reviewer 2 Report

Importance of the study

In Parkinson’s disease (PD), tremor can be one of the most common and troublesome motor symptoms. This tremor is separate from bradykinesia and rigidity because the magnitude of the tremor is not related to dopamine deficiency and does not respond to dopaminergic treatment. There is limited understanding of the pathophysiology of PD tremor and there is ambiguity regarding the involvement of anatomical brain correlates or interconnections between major movement control systems. Particularly in tremor-dominant PD compared to akinetic/rigid subtype, the management of tremor is challenging. Animal studies of locus coeruleus (LC) function are critical for expanding our understanding of the intact LC- noradrenergic (NA) system and for studying the effect of LC damage on tremor in PD.  

To gain insight into the neurochemical basis and anatomical origin of tremor in PD, the authors Pozzi et al, in their present submission, approached the problem, by examining the specific involvement of LC and NA innervation in PD tremor. The LC-NA system has been shown to be relatively preserved in tremor-dominant PD variants. Adopting the reserpinized rat model of PD, the authors selectively lesioned the LC-NA terminals with the neurotoxin (DSP-4) and showed that LC-NA plays a critical role in the development of tremor in PD.

Novelty of the study

The involvement of NA has long been associated in parkinsonian tremor. The concept of LC involvement in parkinsonian tremor and the fine tuning of the LC-NA dynamics including enhanced LC activity as a compensatory response to dopaminergic loss is not new. Damage to the LC-NA system in context to prodromal neurodegenerative changes in PD has also been discussed previously. The anatomical structure and location of LC does not yield easy access for imaging studies on humans. In the past, the classical reserpine model had contributed to a large extent to the role of monoamine system in PD. The present study employing the reserpine rat model for parkinsonian tremor reaffirms the critical role of LC-NA system in tremor development.

Manuscript

The manuscript is well written. Some minor changes may be take into consideration.

Title: “Brain Noradrenergic Innervation Supports the Development of Parkinson’s Tremor: A Study in the Reserpinized Rat Model”. The alternative title could be “Brain Noradrenergic Denervation Ameliorates Parkinson’s Tremor in the Reserpinized Rat Model”

Abstract:

Conforms to the journal guidelines except that in the methods the strain of rats (Sprague Dawley) needs mention.

Introduction: Well written. Tremor subtypes in PD and insight into the origin of the resting and postural components of tremor with interactive role of NA, serotonin and dopamine can be discussed briefly. The involvement of cerebello-thalamo-cortical and basal ganglion-cortical loops can also be touched upon in the introduction.

Materials and Methods:

The animals were divided into 2 groups (i) single reserpine injection (i.p.) (ii) reserpine injection + 14 d later DSP-4 (i.p.). According to treatment, animals were divided in two groups only. The DSP-4 injection alone group is missing. Ideally the study should have included saline-injected control rats as well. 

Quantification of TH- and DBH- neurons and axon densities in 40 µm thick brain slices was performed under fluorescence microscopy. In general, immunohistochemistry with DAB is the preferred and conventional method for quantification. If the authors have made DAB stained slides already those can be utilized for better representation of photomicrographs.   

Results:

Immunofluorescence of NA-LC neurons along with nuclear staining (DAPI, blue) is required for differentiating between cell bodies and fibres. Precise quantification methods like DAB IHC method for measuring the axonal and neuronal densities can also be considered.  

Discussion:

Since, the onset, severity, and progression of tremor are hypothesized to be multifactorial, the role of various brain loci and not solely the LC in PD tremor cannot be ruled out. Further investigations into the origin of tremor subtypes, resting and postural components of tremor as well as factors (cellular activities, cognitive stress) affecting tremor episodes and tremor aggravation might be insightful.

Author Response

Title: “Brain Noradrenergic Innervation Supports the Development of Parkinson’s Tremor: A Study in the Reserpinized Rat Model”. The alternative title could be “Brain Noradrenergic Denervation Ameliorates Parkinson’s Tremor in the Reserpinized Rat Model”

- We thank the reviewer for the suggestion. For the sake of consistency and given the discussion of our findings in terms of “preserved” noradrenergic innervation, we would prefer to keep the original title.  

Abstract: Conforms to the journal guidelines except that in the methods the strain of rats (Sprague Dawley) needs mention

- We included the information in the abstract (page 1, line 51)

Introduction: Well written. Tremor subtypes in PD and insight into the origin of the resting and postural components of tremor with interactive role of NA, serotonin and dopamine can be discussed briefly. The involvement of cerebello-thalamo-cortical and basal ganglion-cortical loops can also be touched upon in the introduction.

- We thank the reviewer for the positive evaluation. As requested, we have now also briefly discussed/implemented the role of other neurotransmitters in tremor pathophysiology. For sake of clarity, we have kept this digression concise (page 2, line 71-74)

Materials and Methods: The animals were divided into 2 groups (i) single reserpine injection (i.p.) (ii) reserpine injection + 14 d later DSP-4 (i.p.). According to treatment, animals were divided in two groups only. The DSP-4 injection alone group is missing. Ideally the study should have included saline-injected control rats as well. Quantification of TH- and DBH- neurons and axon densities in 40 μm thick brain slices was performed under fluorescence microscopy. In general, immunohistochemistry with DAB is the preferred and conventional method for quantification. If the authors have made DAB-stained slides already those can be utilized for better representation of photomicrographs.

- We thank the reviewer for pointing out these aspects that we agree with. Unfortunately, we are unable to incorporate these additional analyses into our work. In the hope that it is still sufficient/acceptable for publication of the article, we have acknowledged these limitations in the discussion (page 5, line 254-259).

Results: Immunofluorescence of NA-LC neurons along with nuclear staining (DAPI, blue) is required for differentiating between cell bodies and fibres. Precise quantification methods like DAB IHC method for measuring the axonal and neuronal densities can also be considered.

- Thank you for pointing out this limitation that we have included in the discussion (page 5, lines 254-259), see also previous comment.

Discussion: Since, the onset, severity, and progression of tremor are hypothesized to be multifactorial, the role of various brain loci and not solely the LC in PD tremor cannot be ruled out. Further investigations into the origin of tremor subtypes, resting and postural components of tremor as well as factors (cellular activities, cognitive stress) affecting tremor episodes and tremor aggravation might be insightful.

- As the reviewer correctly pointed out, there is still much to be investigated in the pathophysiology of tremor and our findings address only one specific topic, namely the role of noradrenergic innervation in the development of PD-related tremor. We agree with the Reviewer that future studies should address the role and interaction of different neurotransmitters and their impact on brain network functioning. We mentioned this at the end of the discussion, hoping to raise interest and foster new studies in this field (page 5, 264-266)

Round 2

Reviewer 2 Report

NA